# Advantages of Porcine Xenograft over Autograft in Sinus Lift: A Randomised Clinical Trial

**DOI:** 10.3390/ma14123439

**Published:** 2021-06-21

**Authors:** Francisco Correia, Daniel Humberto Pozza, Sónia Gouveia, António Campos Felino, Ricardo Faria-Almeida

**Affiliations:** 1Department of Oral Surgery and Periodontology, Faculty of Dental Medicine, University of Porto, 4200-135 Porto, Portugal; antonioccfelino@gmail.com (A.C.F.); rfaperio@gmail.com (R.F.-A.); 2Department of Biomedicine, Faculty of Medicine, University of Porto, 4200-319 Porto, Portugal; dhpozza@gmail.com; 3Department of Histology, Faculty of Nutrition and Food Sciences, University of Porto, 4150-177 Porto, Portugal; 4Institute for Research and Innovation in Health—I3s, University of Porto, 4200-135 Porto, Portugal; 5Department of Electronics, Telecommunications and Informatics (DETI), Institute of Electronics and Informatics Engineering of Aveiro (IEETA), University of Aveiro, 3810-193 Aveiro, Portugal; sonia.gouveia@ua.pt

**Keywords:** bone grafting, sinus augmentation, sinus floor augmentation, bone substitutes, bone regeneration, split mouth randomised clinical trial

## Abstract

This study aimed to compare the performance of intra-oral autologous bone grafts versus porcine xenografts in a two-step lateral window sinus lift. This split-mouth randomised controlled trial sequentially enrolled 12 patients with a 6-month follow-up. For each patient, a simultaneous randomised bilateral maxillary sinus lift was performed and filled with autologous bone from the mandible (control) or a porcine xenograft (test). A bone biopsy sample was collected during the implant placement for histological and histomorphometric analysis. CT scans were performed at the beginning and at the end of the trial to assess radiological evolution. A comparison of initial and six-month CT scans indicated statistically significant increases in bone level for both materials (7.8 ± 2.4 mm for autologous and 8.7 ± 2.2 mm for xenograft, *p* < 0.05), and there were no significant differences between the performance of the two materials over time (*p* = 0.26). The histological analysis showed various stages of the remodelling process and no cells or other signs of inflammation or infection were visible in both groups. The porcine xenografts presented similar results for the studied variables when compared to autologous bone, being a reasonable alternative for a sinus lift.

## 1. Introduction

The maxillary double direction reabsorption, due to centripetal loss of the alveolar bone and sinus pneumatisation, together with thin cortex and low trabecular density, poses a challenge for oral rehabilitation with dental implants [1,2]. This resorptive process is dependent on the number of lost teeth [3] and may lead to a remaining bone height of less than 1 mm [4]. A limited amount of alveolar bone available may result in a lack of primary stability and difficulty in achieving osteointegration. In this context, additional surgical procedures are mandatory to recover bone volume [5,6,7].

Previously or simultaneously to implant placement, maxillary sinus elevations can be performed by lateral osteotomy or maxillary sinus elevation with osteotomes [8,9]. Depending on the risks of morbidity, time and costs of these types of treatments, other options, such as the use of short implants and/or zygomatic implants, should be weighed [10]. Maxillary sinus elevation by lateral osteotomy and bone graft is the preferred treatment when the available bone is <5 mm [5,11,12,13,14]. Additionally, when bone height varies between 3 and 5 mm, simultaneous placement of the standard-length implant is an option [7,11]. The choice of the treatment should be primarily based on anatomy, sinus health, desired bone augmentation, bone dimensions, general health status, smoking, oral hygiene, and preferences [5,13,15].

The ideal bone graft should be biocompatible, osteoinductive, osteoconductive, completely replaced by new bone, capable of maintaining stable grafted volume, with good mechanical properties, derivative from a patient-friendly source, and it should also possess good handling characteristics [10,13,16,17,18,19]. Despite being the gold standard, autologous bone [20,21,22,23,24,25,26,27] presents some limitations such as increased morbidity and limited availability [21]. In this context, several experimental studies on animals and humans have shown that porcine xenografts with 10% collagen exhibit excellent properties in terms of handling, biocompatibility and osteoconductivity, high absorption rate of the xenograft granules and major remodelling by autologous bone after 6 months [28,29,30,31].

A limited number of randomised studies have reported the properties of xenografts based on collagenated corticocancellous porcine bone [32], porcine bone-derived biomaterial [33] and particulate bone substitute of equine origin [34]. The literature, in general, evidences that substitutes and autologous bone exhibit comparable clinical performance, prospecting the successful use of bone substitutes in bone regenerative procedures. Furthermore, several studies demonstrated excellent xenograft biocompatibility with newly formed bone in maxillary sinus augmentation procedures and no evidence of inflammatory process [32,33,34].

This randomised split-mouth controlled clinical trial aimed to compare autologous bone versus porcine xenograft in bilateral sinus lift procedures through clinical, radiological, histological and histomorphometric outcomes. The study focused the properties of Osteobiol MP3^®^, Tecnoss™, Torino, Italy (OMP3) porcine bone xenograft over the above-mentioned evaluation dimensions.

## 2. Materials and Methods

### 2.1. Study Design

This clinical trial was designed in a triple-blind randomised split-mouth and was approved by the Ethics and Research Committee of the Faculty of Dentistry, University of Porto, Portugal, with number 00977 and register at trial.gov with reference NCT01836744. This study followed the CONSORT Statement and World Medical Association Declaration of Helsinki Guidelines [29].

Twelve consecutive patients were included in this study, from March 2013 to June 2017, and received an autologous bone from the mandible and Osteobiol MP3^®^ (OMP3) porcine bone xenograft in each side of the mouth, thus totalling 24 sinuses divided into two groups (autologous and xenograft). In this study, both materials were tested within the same patient to maximise the statistical power in the analysis. Exclusion criteria: irradiated in the head and neck; immunosuppressed or immunocompromised; treated or under intravenous treatment of bisphosphonates; untreated periodontitis; poor oral hygiene and low motivation; uncontrolled diabetes mellitus; pregnant or breastfeeding; psychiatric problems; sinusitis.

### 2.2. Surgical Procedures

The surgical preparation included dental polishing 10 days before, 2 g of amoxicillin 1 h before, and 1 min preceding the beginning of the sinus lift procedure mouthwash with chlorhexidine 0.2%.

All surgeries were performed by the same surgeon, preceded by local anaesthesia (Articaine 4 mg/mL + 0.01 mg/L Artinibsa™, Barcelona, Spain), followed by an incision on the maxillary crest, with two accessory vertical incisions. The detachment and elevation of the flap were performed in total thickness exposing the lateral bone wall of the maxillary sinus [10].

A lateral oval shape osteotomy was performed with a piezoelectric instrument (NSK VarioSurg™, Tokyo, Japan) or with a spherical diamond tip (NSK™, Tokyo, Japan). The Schneider’s membrane was carefully lifted through proper curettes (Hu-Friedy^®^, Chicago, IL, USA). Following the Schneider’s membrane elevation, a randomised opaque sealed sterilized envelope containing the assignment code was opened: (1). test with a cortico-medullar porcine xenograft Osteobiol Mp3^®^ (Tecnoss™, Torino, Italy, granulometry between 600–1000 µm with 10% collagen of type I and III) [29,30]; and (2). control with harvest of intraoral autologous bone graft.

Autologous bone was collected from mandible body or mental symphysis and particulate with a bone mill (ACE™, Pearl St. Brockton, MA, USA). Sinus cavities were filled up to the desired height. The lateral osteotomy was then covered with a collagen membrane (Osteobiol by Tecnoss™, Torino, Italy) and the flaps were sutured with polyamide 4.0 (Supramida™, B Braun, Melsungen, Germany).

The postoperative protocol comprised: Amoxicillin 1 g for a week; chlorhexidine gel 0.2% for 2 weeks; Ibuprofen 400 mg, if pain; soft diet for 2 weeks; ice application within the first 48 h. Additionally, the patients should avoid blowing their noses and using straws; sneezing with their mouth opened; and using any superior removable prosthesis. The sutures were removed after 10 days.

Six months after surgical intervention, a new computer tomography (CT) was made, patients were inquired about treatment preferences and the second surgical stage was performed by the same surgeon under local anaesthesia administration, while the incision and the elevation of the flap were performed in total thickness. The bone collection was performed on both regenerated sides before the use of the first drill of the implant system. The procedure was carried on trans alveolar in the implant sites, with a trephine with an internal diameter of 2 mm and a total diameter of 3 mm (Hager & Meisinger™, Neuss, Germany). Samples were immediately immersed in flasks containing buffered formaldehyde. After bone sample collection, submerged dental implants (Osseo Speed TX™, Astra Tech™, Mölndal, Sweden) were placed following the manufacturer’s recommendations. The flaps were sutured with polyamide 4.0 suture (Supramida™, B Braun™, Melsungen, Germany). After ten days, the sutures were removed, and eventual postoperative complications were clinically evaluated and registered.

### 2.3. Radiological Analysis

Radiological analysis (computer tomography—CT) comprised hard tissue levels at baseline (at the time of the patient recruitment) and 6 months. Hard tissue levels were quantified from each CT using the Blue Sky Plan^®^ software (Blue Sky Bio^®^, LL Grayslake, IL, USA), and the distance between the coronal bone level and the apical bone level was measured in the region where the regenerative procedure was performed. Ten consecutive sagittal cuts with 1 mm interval were obtained and measured at each CT side (Figure 1).

### 2.4. Histology Processing and Histomorphometric Analysis

Histological samples were processed and stained (hematoxylin and eosin—H&E) following the procedures described in previous literature studies [28,34,35]. The 10 slides from the most central and representative part of the anatomical piece were used for the histological and histomorphometric analysis. 

The photographs were taken in an optical microscope (Zeiss Axisoskop 40™, Carl Zeiss, Berlin, Germany) at a magnification of 50× for the histomorphometric measurements through proper software (ImageJ™ 1.5.0 by National Institutes of Health, Bethesda, MD, USA). Total area, hard tissue area (bone and graft), and soft tissues (collagen, fibroblasts, blood vessel, adipocytes, and void spaces) were measured in pixels. These results were converted in percentages for the statistical analysis. Photographs were taken at magnifications of 200× and 400× for additional histological analysis by a “blind” dentist with expertise in the area.

### 2.5. Statistical Analysis

The statistical analysis, data processing and visualization were performed in Microsoft Excel™ 16.10 and IBM™ SPSS™ (25.0, IBM Corp., Armonk, NY, USA) with a significance level of 5% by another blind and independent expert assessor. The statistical comparisons were based on paired t-test, one-way ANOVA or two-way ANOVA, depending on the purpose of the analyses. When needed, the data normality was tested via the Shapiro-Wilk test. In all cases, normality was not rejected and the above-mentioned parametric statistical comparisons were applied. Finally, numerical results throughout the text are presented in the mean ± standard deviation (SD) format.

## 3. Results

### 3.1. Patient and Intervention Characteristics

A total of 24 sinus elevations were evaluated from 12 patients (six males and six females) with an average age of 59.7 ± 8.7 years old. The number of non, light, heavy and former smokers was six, three, one, and two, respectively. Most of the patients exhibited up to one pathology (9 out of 12) and were taking at least one medication (11 out of 12).

Autologous bone was harvested from the mandible branch in 83.3%, and from the chin in only 16.7% of the cases. 11 out of 12 patients (91.6%) reported that “Neither / both procedures were equally good” and only one patient (8.3%) indicated a preference for “The enlarged site with the bone substitute” (Table 1). There were no major intra- or post-surgery complications, except for a small perforation of the Schneider’s membrane (<2 mm) in 5 out of the 24 sinuses that were treated with a resorbable collagen membrane (Table 1). 

### 3.2. Radiological Findings

Baseline tissue heights were not significantly different for autologous and xenograft materials (3.20 ± 0.93 and 3.06 ± 1.13 mm, *p* = 0.69) ensuring similarity for both groups (Table 1). The tissue height gain presented values ranging from 3.7 to 12.5 mm for autologous graft and from 5.4 to 12.5 mm for xenograft allowing placement of dental implants from 9 to 11 mm in length. The paired analysis between the baseline and the final tissue height values also showed no significant correlation between observation times (baseline and 6 months after intervention) for both materials (r = 0.149, *p* > 0.05 and r = 0.298, *p* > 0.05, respectively for autologous graft and xenograft). 

In 75% of the cases, a greater increase in tissue height was observed with the use of the xenograft. Although the tissue height increase was slightly higher for the xenograft than for the autologous graft, the difference between materials performance did not reach significance (7.81 ± 2.34 and 8.71 ± 2.15 mm, *p* = 0.26). Finally, two-way ANOVA on the interaction between tissue height over time and materials showed a significant effect on time (*p* < 0.001), but no significant effect on the material (*p* = 0.240) and no significant interaction between time and material (*p* = 0.261). 

### 3.3. Histological Findings

Histological examination of the 240 slides showed various stages of the remodelling process with both the autologous bone and the xenograft integrated into the newly formed bone tissue, being partially involved by vascularised tissue. No signs of inflammation or infection were observed in any of the samples.

The evaluation of the xenograft biopsies demonstrated bone remodelling with osteoclasts, new bone apposition by osteoblasts and, consequently, osteocytes. These highly vascularised areas of active bone formation alternated with zones of mature bone. It was also visible that both grafts’ particles were well incorporated into the newly formed bone tissue. Figure 2A (control) and Figure 2B (test) show areas of soft tissues in interface with hard tissue areas.

In the experimental group, it was possible to observe active zones with residual xenograft particles well incorporated into the newly formed bone and osteoclasts (Figure 3B,D). In the control group, the autologous bone graft was almost completely resorbed. The histological analysis demonstrated only a few remaining particles fully integrated with the newly formed vital bone (Figure 3A,C). The medullary spaces, between the bone and graft particles, were filled by soft tissue rich in blood vessels. 

Figure 3C (autologous graft) shows the bone vitality represented by osteoclastic reab-sorption of the autologous bone graft and also apposition of new bone with osteoblasts. Figure 3D (xenograft) shows soft tissue partially surrounding graft fragments and mature bone fragments, in addition to the presence of osteoclasts and osteoblasts. Xenograft particles have irregular lacunae due to osteoclast resorption and immature bone deposition process by osteoblasts. The presence of osteocytes, at 6 months, demonstrated maturation of the regenerated bone. The residual graft particles were detected and identified by their atypical colour, structure and the presence of empty lacunae, being in direct contact with the vital bone.

### 3.4. Histomorphometric Findings

The results of histomorphometric data, represented in terms of total hard tissue volume (THT) and total soft tissue volume (TST), were similar for both groups (Table 2).

The mean THT for the autologous bone side was 57.31 ± 2.91%, and for the xenograft side: 56.01 ± 2.86% (*p* = 0.376), being patient-dependent. In this context, the minimum THT (49.67 vs. 44.19%) and maximum THT (64.07 vs. 64.19%) values were very close and occurred in the same patient.

Standard deviation (SD) associated with the autologous bone ranges from 1.29% to 7.03%, and the average SD in the sample is 2.91%. The relatively low SD values indicate that the dispersion of autologous bone ratio values obtained from several slides to the corresponding mean ratio is relatively low, thus supporting that the mean value is a reasonable indicator.

The correlation between the THT values for both grafts (Table 2) was positive, although not statistically significant (r = 0.546, *p* = 0.066). The positive correlation points out that patients exhibit higher (or lower) THT values concordantly for both grafts, which suggests that results may also depend on the patient rather than only the graft. On the one hand, in the two-way ANOVA analysis, there was no significant effect of the graft type on the histomorphometric performance (*p* = 0.376). On the other hand, a significant effect of the patient was observed (*p* = 0.029), thus supporting that the performance depends on the patient (and consequently on their individual characteristics). Finally, there was a significant interaction effect between material and patient effects (*p* < 0.001), showing that the histomorphometric result of the same material also depends on the patient (Figure 4). The overlap of the 95% confidence intervals associated with both materials for each patient is observed in seven patients (i.e., there are no differences between the materials). Further, in 3 of the 12 cases, the average THT was higher for the autologous bone procedure, and in 2 of the 12 cases, there was an inverse behaviour (higher average THT for the xenograft). 

## 4. Discussion

This research demonstrated that, after six months of lateral osteotomy sinus lift, both porcine xenograft and autogenous bone presented similar results in terms of radiologic, clinical, histological, and histomorphometric variables. These findings are supported by the fact that this is a double-blind randomised split-mouth clinical trial type IV in which patients were carefully chosen to ensure adequate comparisons after the blinding se-quence and randomisation of treatments [36,37].

The results of the histological evaluation showed the existence of various stages of bone remodelling, both at tissue and cellular levels. Interestingly, the turnover was more evident in the xenograft side, probably due to intrinsic high biocompatibility and oste-oconductive properties. Corroborating these findings, Ramirez-Fernandez, MP et al. [38] also observed substantial reabsorption of the grafted material, after 9 months. Ad-ditionally, it was possible to visualize an apposition of newly lamellar bone containing osteocytes trapped in bone gaps. These highly vascularised, active bone-forming zones alternated with mature bone tissue zones. The high rate of resorption of Osteobiol MP3^®^ granules, presenting fragments surrounded by autologous bone and recently vascular-ised tissues, and demonstrating excellent biocompatibility and osteoconductivity, was also previously reported [29,30]. Besides the cortico-medullar porcine origin, the addition of collagen gel types I and III to the matrix facilitated the OMP3 manipulation, allowing excellent biocompatibility properties, as demonstrated in the present and previous studies [28,29,30,31,39].

However, one of the main disadvantages described in the histologic studies for other biomaterials such as bovine xenografts, over time, is that its resorption is slow and, in many cases, incomplete, even after many year [40]. This fact is considered a disadvantage since most of the bovine graft volume is not replaced by newly vital bone [18], as demonstrated for OMP3 in the present study. The better replacement of xenografts by vital bone is probably due to the collagen influence on cellular and molecular activity, inducing the adhesion of osteoclasts to the surface of the biomaterial [29,38]. This action of collagen is of great importance since the use of a porcine bone graft has led to either minimal or non-existent resorption of the graft at 6 months [33,34].

The transplanted autologous bone also presented high biological rates, although inferior to OMP3. These histological results were expected at six months and it is believed that the remaining grafts may be a consequence of the larger size of the shredded autologous bone particles, that were taking longer to be remodelled. This is in accordance with a recent meta-analysis [23] that reported a significantly higher proportion of mineralized autog-enous bone during the early healing phase when compared to various bone substitutes used alone or in combination with autogenous bones. However, after a healing period of more than 9 months, no significant differences were found between the different treat-ment modalities.

A remarkable growth of new bone around particles was observed at high magnification on histological analysis, with signs of active resorption of bone grafts, both autologous and xenograft [27,29,38]. The bone/soft tissues proportion was similar for both the autologous bone and xenograft groups and close to values obtained in previous studies reported in the literature [28,32,41]. However, in a case series [30], the total values of formed bone were slightly lower (42.3 ± 9.7%) than those reported in the present clinical trial. Addi-tionaly, it was demonstrated, in a Bayesian network meta-analysis [42], trough histo-morphometry analisys, that the range of new bone varied from 10.41% to 50.23%. On the other hand, studies are reporting higher values for bone percentage when compared to soft tissues [43]. These higher results may be related to the technique per se, as well as, the way that the histomorphometric analysis was performed.

Additionally, and in accordance with previous studies, it was difficult to distinguish the interface between hard tissues of residual bone and grafts [29,41]. Thus, the residual graft volumes were not individually measured, since there was very reduced remaining that would add no practical information. It is also important to mention that the goal of the surgical procedure underlying this research was to achieve around 8 mm of bone height for the placement of standard-length implants. Similar results were previously found in a case series study [30] that also used OMP3 and reached a bone height with an average of 9.6 ± 3.8 mm. The results of this study were also similar to those obtained in the Bayesian network study in terms of vertical bone gain [44] and also in accordance with a me-ta-analysis [45] for the groups with follow up between 4.5 and 9 months.

A significant patient effect was observed as a function of its characteristics regardless of material, as well as, a significant interaction effect between patient and material, pointing out that the performance of the material depends on the patient. This difference may be due to the location of the collection and the size of the histological sample collected, which contained a larger amount of residual bone present since the bone crest is the site of the largest source of mesenchymal stem cells involved in angiogenesis [46,47]. Other hy-potheses were explored and statistical analysis was performed in order to relate this factor to others involved, such as age or gender: “Older women exhibit poorer bone regener-ation capacity after sinus enlargement than young patients” [47] or with a perforation of the Schneider membrane [48], however, no conclusions can be drawn due to the limitation of the small sample size.

Another important statistical conclusion was that the performance of the two materials over time was fairly similar, with no significant material effect at 5% level and interaction between time and material. This conclusion supports that the value of bone gain does not depend largely on the initial value, and suggests that there are other determinants of bone gain at the end of the intervention, such as the amount of graft placed, the anatomical shape of the maxillary sinus, and/or the surgeon’s ability to properly lift the Schneider membrane. From a clinical point of view, a greater increase is expected with the use of biomaterials than with the use of autologous bone as the limitation on the amount of graft available and harvestable is eliminated.

Clinical results, including the absence of infection or inflammation, were histologically confirmed since none of the samples had inflammatory cells, tissue alterations nor foreign body reactions, being in accordance with previous studies that also used OMP3 [28,43]. This is of extreme importance since other biomaterials such as bovine cortical bone, cal-cium sulphate, coral hydroxyapatite, and bioactive glass can present, in some cases, in-flammatory complications [43]. For example, a recent randomised split-mouth clinical trial [49], that used biphasic calcium phosphate as graft material in a sinus lift procedure, evidence of a mild and chronic inflammatory infiltrate was histologically demonstrated.

Both materials were found to be suitable in a two-step lateral window sinus lift sinus, with advantages for porcine xenograft over autograft. However, the limitations of this study included the small sample size, given the heterogeneity observed among patients, which may be carefully evaluated for statistical inference. Our results highlight that the dif-ference between materials is smaller than the variability of patients (large heterogeneity e.g., in smoking history and other pathologies) which may have influenced the outcome of the experimentation.

## 5. Conclusions

Based on the histological and histomorphometric as well as clinical and radiological results, it can be concluded that the OMP3 xenograft is a valid alternative to the use of autologous bone grafts in lateral maxillary sinus osteotomy procedures due to its excellent osteoconductivity and biocompatibility, as well as an absence of foreign body reactions or infections.

## Figures and Tables

**Figure 1 materials-14-03439-f001:**
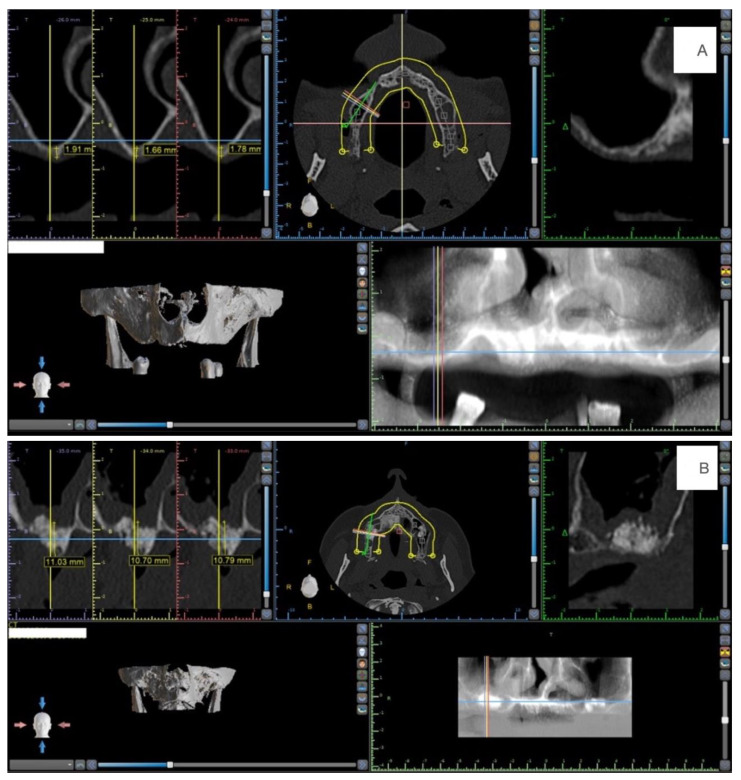
(**A**) Example of the measurements at the baseline. (**B**) Example of the measurements at the 6 months.

**Figure 2 materials-14-03439-f002:**
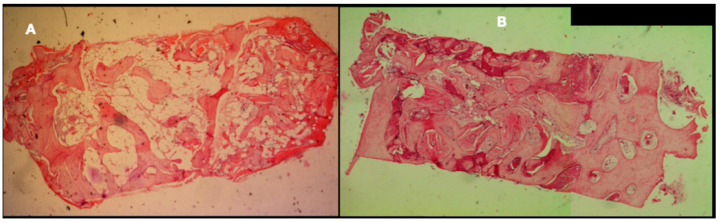
(**A**) Example of an autograft-regenerated sinus side—H&E staining—50×. (**B**) Example of a xenograft-regenerated sinus side—H&E staining—50×.

**Figure 3 materials-14-03439-f003:**
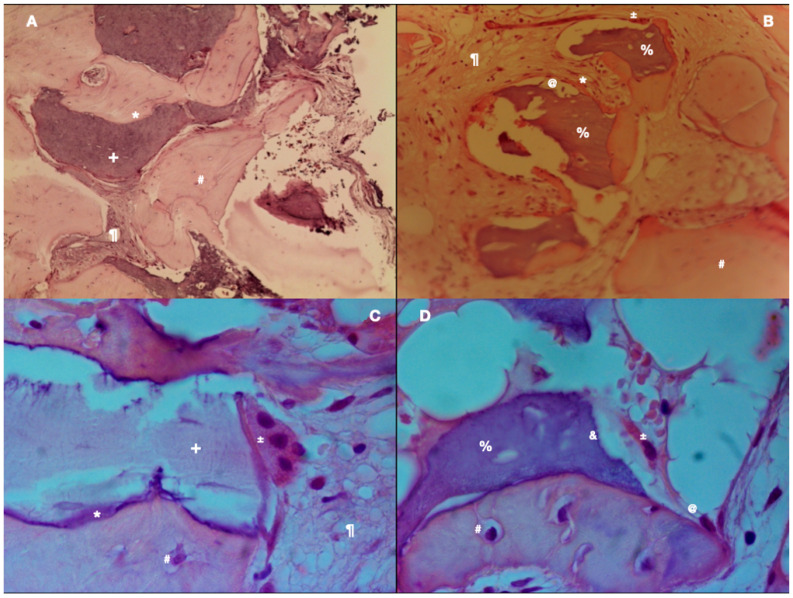
(**A**) Example of autologous bone graft—H&E coloration—200×. (**B**) Example of xenograft particles—H&E staining—200×. (**C**) Autologous bone graft sample—H&E staining—400×. (**D**) Example of xenograft particle in high magnification—H&E staining—400×. Legend: * immature bone, + autologous bone graft, # osteocyte, % xenograft, ¶ soft tissue, @ osteoblast, ± osteoclast, & Howship lacunae.

**Figure 4 materials-14-03439-f004:**
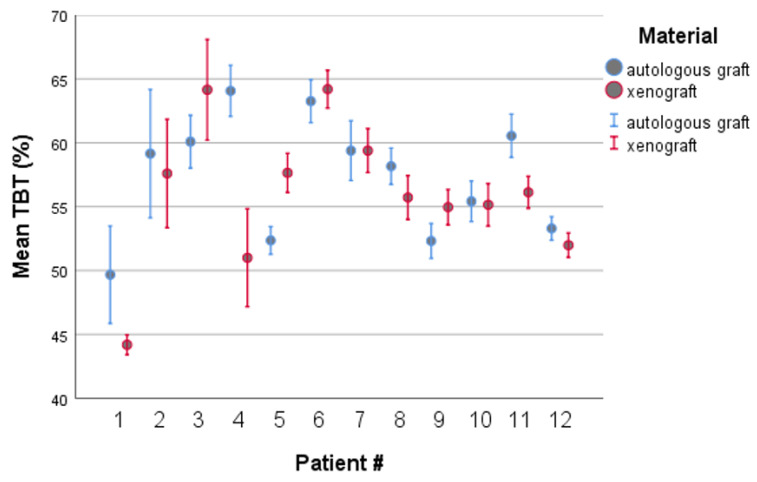
Clustered 95% confidence intervals for average THT by patient and by material, obtained from 10 slides in each condition.

**Table 1 materials-14-03439-t001:** Clinical observation and radiological evaluation.

			Hard Tissue Levels (mm)					
ID	Gender	Age	**ABG**			**Xenograft**		
			Baseline Bone Height (Mean)	Mean 6 Months	Diferences mm (6 Months—Baseline)	Baseline Bone Height (Mean)	Mean 6 Months	Diferences mm (6 Months—Baseline)
**1**	F	63.74	2.10	11.30	9.2	4.80	10.15	5.4
**2**	F	42.23	2.60	11.05	8.5	2.85	12.00	9.2
**3**	M	58.66	3.15	6.80	3.7	2.55	9.65	7.1
**4**	M	63.70	4.70	10.15	5.5	2.85	11.55	8.7
**5**	F	67.83	3.30	9.90	6.6	3.60	10.40	6.8
**6**	M	75.04	1.30	8.83	7.5	1.20	9.83	8.6
**7**	M	59.56	4.10	16.56	12.5	3.91	15.11	11.2
**8**	M	54.14	2.59	11.98	9.4	1.54	12.40	10.9
**9**	M	65.54	3.43	9.86	6.4	2.50	15.00	12.5
**10**	F	61.05	3.67	13.13	9.5	4.90	12.44	7.5
**11**	F	55.48	3.88	9.87	6.0	2.67	9.21	6.5
**12**	F	49.49	3.50	12.63	9.1	3.30	13.40	10.1
**Total**	M-6 F-6	$\bar{x}-$ 59.70	3.20 ± 0.93	11.02 ± 2.45	7.81 ± 2.34	3.06 ± 1.13	11.76 ± 2.01	8.71 ± 2.15

Legend: ID—identification code, mean ± standard deviation, $\bar{x}$—mean, M—male, F—female, ABG—Autologous graft.

**Table 2 materials-14-03439-t002:** Histomorphometric results: mean and standard deviation (SD) of 10 slides per patient side (ID) according to the group in terms of % of total hard tissue and total soft tissue.

ID	Total Hard Tissue Volume (THT%)				Total Soft Tissue (TST%)			
	Autologous		Xenograft		Autologous		Xenograft	
	Mean	SD	Mean	SD	Mean	SD	Mean	SD
1	49.67	5.33	44.19	1.08	50.33	5.33	55.81	1.08
2	59.15	7.03	57.60	5.94	40.85	7.03	42.40	5.94
3	60.09	2.90	64.15	5.51	39.91	2.90	35.85	5.51
4	64.07	2.79	51.00	5.36	35.93	2.79	49.00	5.36
5	52.35	1.52	57.65	2.14	47.65	1.52	42.35	2.14
6	63.26	2.35	64.19	2.07	36.74	2.35	35.81	2.07
7	59.39	3.27	59.39	2.41	40.61	3.27	40.61	2.41
8	58.16	1.99	55.72	2.40	41.84	1.99	44.28	2.40
9	52.31	1.91	54.96	1.94	47.69	1.91	45.04	1.94
10	55.42	2.22	55.15	2.33	44.58	2.22	44.85	2.33
11	60.55	2.37	56.12	1.75	39.45	2.37	43.88	1.75
12	53.29	1.29	51.98	1.33	46.71	1.29	48.02	1.33
mean	57.31	2.91	56.01	2.86	42.69	2.91	43.99	2.86

## Data Availability

Not applicable.

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
