# Peer review of "Advantages of Porcine Xenograft over Autograft in Sinus Lift: A Randomised Clinical Trial"

_materials, 2021, doi:10.3390/ma14123439_

Round 1

Reviewer 1 Report

I congratulate for the authors of this study! The manuscript of "Advantages of porcine xenograft over autograft in sinus lift: a 2 randomised clinical trial" is a very clinical orientated study, which aims to compare the performance of intraoral autologous bone graft versus porcine xenograft (Osteobiol MP3) in a two-step lateral window sinus lift.

The following minor comments and suggestions shall be considered:

The content of Line 74-80 was not properly placed in the manuscript since the content describes the previously published study designs and places the actual study design in the literature. According to the formal requirements it shall be placed in the introduction part.

In line 108 "the flaps were sutured" is written, but in the description of post operative protocol the removal of the sutures is missing. 

In the manuscript CT was indicated as the imaging modality in all sections. The brand of radiological equipment, and its manufacturing country shall be specified. If other specifications of the acquisitions are available, it would incresase the reproducibility of the imaging method.

Pre- and postoperative CT images on which the measurements were carried out might be illustrative in the manuscript and facilitate the smooth reading and interpretiation of the reader.

In Line 124 "(CT)" does refer to a possible abbreviation or to a product? 

In line 124 the "baseline" must be clarified, hence the reader can interpret with ease when the first CT scan was carried out.

In line 126-127 "coronal bone level and the apical bone level (mm) was measured". It is needed to be explained in more detail (e.g. on which CT plane was it measured?)  

In line 127 "(mm)" in the sentence shall be explained why (mm) is written there. 

If only one staining method was used it shall be indicated in the section of 2.4 Histology processing and histomorphometric analysis first.

The method for generating the data reported by the patients of Table 1 ("Treatment Preferences Reported by the Patient") is missing from Materials and Methods section. It needs to be indicated prior to the Results section.

In line 168 the wording of "observation times" must be clarified. 

There is no Figure 3 in the manuscript.

In Line 262-263 an abbreviation of Osteobiol MP3 (OMP3) was indicated. The abbreviation shall be placed next to the first appearance of the expression (Line 71).

Typing error in Line 294: "tissue[42]." A space needs to be inserted: tissue [42].

In the References section the references need to be reedited according to the formal requirements (https://www.mdpi.com/authors/references). For instance, at this stage, in the manuscript some of the name of journals are abbreviated some of them are not. Furthermore the most recent cited reference was published in 2018 and it is the only one from this year. The other references are older than 2018. It might be appropriate if the inclusion of further citations of newer publications, including metaanalysis in this field would be considered. Such as Trimmel, B.;, Gede, N.; Hegyi, P.; Szakács, Z.; Mezey, G.A.; Varga, E.; Kivovics, M.; Hanák, L.; Rumbus, Z.; Szabó, G. Relative performance of various biomaterials used for maxillary sinus augmentation: A Bayesian network meta-analysis. Clin. Oral. Implants. Res. 2021, 32, 135-153. DOI:10.1111/clr.13690. 

Author Response

Thank you very much for the excellent suggestions and criticisms. We carefully analysed the issues raised in the comments to improve the manuscript. The responses to the reviewer’s comments are presented in the following text (in a point-by-point fashion) and the manuscript alterations are highlighted in yellow colour in the document of the manuscript text.

Comments and Suggestions for Authors

I congratulate for the authors of this study! The manuscript of "Advantages of porcine xenograft over autograft in sinus lift: a 2 randomised clinical trial" is a very clinical orientated study, which aims to compare the performance of intraoral autologous bone graft versus porcine xenograft (Osteobiol MP3) in a two-step lateral window sinus lift.

Response: Thank you for the comment that will help to improve the paper. We have reviewed the English.

The following minor comments and suggestions shall be considered:

The content of Line 74-80 was not properly placed in the manuscript since the content describes the previously published study designs and places the actual study design in the literature. According to the formal requirements it shall be placed in the introduction part.

Response: Thank you for the comment, we change according your suggestion.

“A limited number of randomised studies have reported the properties of xenograft based on collagenated corticocancellous porcine bone [1], porcine bone-derived biomaterial [2] and particulate bone substitute of equine origin [3]. The literature, in general, evidences that substitutes and autologous bone exhibit comparable clinical performance, pro-specting the successful use of bone substitutes in bone regenerative procedures. Fur-thermore, several studies demonstrated excellent xenograft biocompatibility with newly formed bone in maxillary sinus augmentation procedures and no evidence of inflam-matory process [1-3].”

In line 108 "the flaps were sutured" is written, but in the description of post operative protocol the removal of the sutures is missing.

Response: Thank you for the comment, we added the missing information.

“The sutures were removed after 10 days.”

In the manuscript CT was indicated as the imaging modality in all sections. The brand of radiological equipment, and its manufacturing country shall be specified. If other specifications of the acquisitions are available, it would incresase the reproducibility of the imaging method.

Pre- and postoperative CT images on which the measurements were carried out might be illustrative in the manuscript and facilitate the smooth reading and interpretiation of the reader.

Response: Thank you for the comment. A figure was added to the main document and more information.

“10 consecutive sagittal cuts with 1mm interval were obtained and measured at each CT side (figure 1).

Figure 1A - Example of the measurements at the baseline.

Figure 1B - Example of the measurements at the 6 months.

In Line 124 "(CT)" does refer to a possible abbreviation or to a product?

Response: Thank you for the comment. We have added the missing information.

“computer tomography (CT)”

In line 124 the "baseline" must be clarified, hence the reader can interpret with ease when the first CT scan was carried out.

Response: Thank you for the comment. We have added the missing information.

“baseline (at the time of the patient recruitment)”

In line 126-127 "coronal bone level and the apical bone level (mm) was measured". It is needed to be explained in more detail (e.g. on which CT plane was it measured?)

Response: Thank you for the comment. We added to the main document more information.

“10 consecutive sagittal cuts with 1mm interval were obtained and measured at each CT side (figure 1).

In line 127 "(mm)" in the sentence shall be explained why (mm) is written there.

Response: Thank you for the comment we have corrected the typo.

If only one staining method was used it shall be indicated in the section of 2.4 Histology processing and histomorphometric analysis first.

Response: Thank you for the comment we added the missing information.

“(hematoxylin and eosin - H&E)”

The method for generating the data reported by the patients of Table 1 ("Treatment Preferences Reported by the Patient") is missing from Materials and Methods section. It needs to be indicated prior to the Results section.

Response: Thank you for the comment. We have add more information.

“ patients were inquired about treatment preferences”

In line 168 the wording of "observation times" must be clarified. There is no Figure 3 in the manuscript.

Response: Thank you for the comment we add more information in order to clarified the sentence.

“observation times (baseline and 6 months after intervention)”

Figure 4 - Clustered 95% confidence intervals for average THT by patient and by material, obtained from 10 slides in each condition.

In Line 262-263 an abbreviation of Osteobiol MP3 (OMP3) was indicated. The abbreviation shall be placed next to the first appearance of the expression (Line 71).

Response: Thank you for the comment. We have modified it.

Typing error in Line 294: "tissue[42]." A space needs to be inserted: tissue [42].

Response: Thank you for the comment. We have correctted the typo.

In the References section the references need to be reedited according to the formal requirements (https://www.mdpi.com/authors/references). For instance, at this stage, in the manuscript some of the name of journals are abbreviated some of them are not. Furthermore the most recent cited reference was published in 2018 and it is the only one from this year. The other references are older than 2018. It might be appropriate if the inclusion of further citations of newer publications, including metaanalysis in this field would be considered. Such as Trimmel, B.;, Gede, N.; Hegyi, P.; Szakács, Z.; Mezey, G.A.; Varga, E.; Kivovics, M.; Hanák, L.; Rumbus, Z.; Szabó, G. Relative performance of various biomaterials used for maxillary sinus augmentation: A Bayesian network meta- analysis. Clin. Oral. Implants. Res. 2021, 32, 135-153. DOI:10.1111/clr.13690.

Response: Thank you for the comment. We have add new references.

Reviewer 2 Report

Thank you for your submission.

I recommend additional contents for this draft as follows:

  1. It would be good to show additional information about included patients. In particular, although it is written in the text that there is no statistically significant difference between the two groups in the height of residual bone in patients before surgery, information on the baseline bone heights of all patients is required. Also, please provide the patients' age, gender, amount of graft material, and post-op bone height(immediately and after 6 months).

  1. It was said that a CT was taken after surgery, but the results are summarized only in a table. Please attach preoperative and postoperative radiographs.

Author Response

Dear Editor and Reviewers,

Thank you very much for the excellent suggestions and criticisms. We carefully analysed the issues raised in the comments to improve the manuscript. The responses to the reviewer’s comments are presented in the following text (in a point-by-point fashion) and the manuscript alterations are highlighted in yellow colour in the document of the manuscript text. We have reviewed the English.

The following minor comments and suggestions shall be considered:

  1. It would be good to show additional information about included patients. In particular, although it is written in the text that there is no statistically significant difference between the two groups in the height of residual bone in patients before surgery, information on the baseline bone heights of all patients is required. Also, please provide the patients' age, gender, amount of graft material, and post-op bone height(immediately and after 6 months).

Response: Thank you for the comment, we have added the missing information and redid the table 1.

Hard tissue levels (mm)

ID

Gender

Age

ABG

Xenograft

Baseline

Bone Height (Mean)

Mean

6 months

Diferences mm (6 months – baseline)

Baseline

Bone Height (Mean)

Mean

6 months

Diferences mm (6 months – baseline)

1

F

63,74

2,10

11,30

9,2

4,80

10,15

5,4

2

F

42,23

2,60

11,05

8,5

2,85

12,00

9,2

3

M

58,66

3,15

6,80

3,7

2,55

9,65

7,1

4

M

63,70

4,70

10,15

5,5

2,85

11,55

8,7

5

F

67,83

3,30

9,90

6,6

3,60

10,40

6,8

6

M

75,04

1,30

8,83

7,5

1,20

9,83

8,6

7

M

59,56

4,10

16,56

12,5

3,91

15,11

11,2

8

M

54,14

2,59

11,98

9,4

1,54

12,40

10,9

9

M

65,54

3,43

9,86

6,4

2,50

15,00

12,5

10

F

61,05

3,67

13,13

9,5

4,90

12,44

7,5

11

F

55,48

3,88

9,87

6,0

2,67

9,21

6,5

12

F

49,49

3,50

12,63

9,1

3,30

13,40

10,1

Total

M-6

F-6

59,70

3,20 ± 0,93

11,02 ± 2,45

7,81 ± 2,34

3,06 ± 1,13

11,76 ± 2,01

8,71 ± 2,15

Legend:

ID – identification code

    mean ± standard deviation

x Ě…- mean

M – male

F- female

ABG – Autologous graft

  1. It was said that a CT was taken after surgery, but the results are summarized only in a table. Please attach preoperative and postoperative radiographs.

Response: Thank you for the comment. A figure was added to the main document, as well as, additional information.

Figure 1A - Example of the measurements at the baseline

Figure 1B - Example of the measurements at the 6 months
